# Impact of Aquatic-Based Physical Exercise Programs on Risk Markers of Cardiometabolic Diseases in Older People: A Study Protocol for Randomized-Controlled Trials

**DOI:** 10.3390/ijerph17228678

**Published:** 2020-11-23

**Authors:** José Pedro Ferreira, Ana Teixeira, João Serrano, Carlos Farinha, Hélder Santos, Fernanda M. Silva, Márcio Cascante-Rusenhack, Paulo Luís

**Affiliations:** 1FCDEF-UC, CIDAF, University of Coimbra, 3040-248 Coimbra, Portugal; ateixeira@fcdef.uc.pt (A.T.); cmnfarinha@gmail.com (C.F.); geral.fernandasilva@gmail.com (F.M.S.); marciocascante@gmail.com (M.C.-R.); 2Sport, Health & Exercise Research Unit (SHERU), Polytechnic Institute of Castelo Branco, 6000-266 Castelo Branco, Portugal; j.serrano@ipcb.pt; 3Coimbra College of Health Technology—IPC, 3046-854 Coimbra, Portugal; heldersantos98@gmail.com; 4School of Physical Education and Sports, University of Costa Rica (UCR), San José 11501-2060, Costa Rica; 5Municipality of Sertã, 6100-738 Sertã, Portugal; pauloluis@cm-serta.pt

**Keywords:** exercise, aquatic-based, hydro gymnastics, elderly, cardiometabolic diseases

## Abstract

Cardiometabolic diseases are one of the primary causes of mortality and morbidity worldwide and sedentary lifestyles are contributing factors to these pathologies. Physical exercise has been recognized as an important tool in the prevention and treatment of these diseases. However, there are still some doubts about the efficacy of certain type of physical exercise programs for older participants. The main goal of this study is to assess the impact of different aquatic-based physical exercise programs on risk markers of cardiometabolic diseases in older people. The study group will consist of non-institutionalized individuals, within the age group of 65 or older. The sample will be randomly divided into four groups, three experimental groups (EG) and one control group (CG). Participants from the EGs will be exposed to three physical aquatic-based exercise programs for a period of 28 weeks (continuous aerobic, interval aerobic and combined). The evaluated parameters include anthropometry, physical functions, mental health, cognitive function, carotid arteries intima-media thickness, heart rate variability and biochemical markers. The results will allow an interpretation of the impact of different aquatic-based physical exercise programs on cardiometabolic diseases markers and can also be used as a tool for professionals to prescribe adequate and more efficient physical exercise programs.

## 1. Introduction

Ageing is considered a progressive and inevitable phenomenon, where the reduction in various physical and metabolic functions occurs [1]. As they age, humans become increasingly sensitive to certain pathologies such as cardiometabolic diseases. These diseases are considered to be the main cause of mortality and morbidity worldwide and have attracted great interest in various fields of scientific investigation. The World Health Organization (WHO) reports that cardiometabolic diseases are responsible for 63% of 57 million annual deaths, with a significant correlation of 6 to 10% of these deaths due to physical inactivity [2].

Nowadays, more than 60% of the elderly population live a sedentary lifestyle [3]. This contributes to physical and mental frailty. Physical frailty is characterized by a reduction in physical activity levels, unintended weight loss, fatigue, reduction in handgrip strength, and a decrease in walking speed [4]. The same authors characterize cognitive frailty as the decline in all functional mental abilities that regulate the lifestyle of an individual. These mental activities range from simple to complex. Both types of frailty are associated to the process of physical and mental decline, with a deterioration in physical and mental capabilities that may lead to a serious decrease in health conditions, loss of autonomy, institutionalization and mortality [5]. To counteract sedentary lifestyles, WHO has developed strategies for a more active and healthy society. These strategies include the creation of guidelines that state the frequency, duration, intensity, type and amount of physical activity that are recommended for different age groups. Such guidelines target the prevention of non-transmissible chronic diseases [6]. Simultaneously, there has been an increase in research using a set of variables that are related to these types of pathologies, such as body composition [7], functional fitness [8], and variables related to the cardiovascular [9], cognitive [4] and immune systems [10].

Physical exercise is considered an effective tool during the process of ageing, as it helps to stabilize the loss of physical and metabolic capacities, mitigates the overall progress of ageing and enables more autonomy. Various studies indicate that regular physical exercise plays an important role in the prevention and treatment of cardiometabolic and cognitive diseases [10,11,12,13,14].

Different studies have been conducted recently to verify the efficacy of various physical exercise programs (aerobic, muscular strength and combined) on variables related to these pathologies. Such studies have analyzed the impact of exercise on cognition, cardiovascular, metabolic, immunological, functional and mental levels [15,16,17,18]. However, most of these studies have focused on land-based exercise and have not involved specific aquatic exercise programs, which are popular and successful among older participants.

The present investigation and its specific experimental design using different types of aquatic-based exercise programs aims to assess the impact of different aquatic-based physical exercise programs on risk markers of cardiometabolic diseases in older people.

## 2. Materials and Methods

### 2.1. Design

This investigation will be based on a randomized intervention, using three different aquatic-based exercise programs conducted in parallel for 28 weeks. The three programs include a continuous aerobic program, an interval aerobic program and a combined (aerobic and muscular strength) program. Variables related to cardiometabolic diseases will be evaluated, i.e., cardiovascular, mental health, and cognitive variables and biochemical markers.

All exercise programs will be conducted at the Piscina Muncipal da Sertã, two times a week (non-consecutive days), with 45-min sessions. All variables related to cardiometabolic diseases will be evaluated at two specific assessment moments: M1, before implementation of the physical exercise programs (baseline) and M2, after the conclusion of the 28 weeks aquatic-based physical exercise programs (post-intervention).

Each evaluation will be divided into three phases: Phase 1—Anthropometry evaluation, physical functions, cognitive and mental health levels; Phase 2—Evaluation of carotid arteries’ intima-media thickness and heart rate variability; and Phase 3—Evaluation of biochemical markers.

The blood collection for biochemical analysis will be conducted by a specialized and certified lab. The measurement of the carotid arteries intima-media thickness will be conducted by a specialist in the area of cardiology. All the remaining data will be collected and organized by the research team members.

### 2.2. Participants

The participants were recruited in the central area of Portugal, more specifically in the region of Sertã. The sample was recruited by the non-probabilistic method for convenience. One hundred and fifty individuals from the community were personally invited, but only 102 agreed to participate in the study (mean age of 72.32 ± 5.2 and BMI 29.47 ± 4.85). Participants will be selected according to the following inclusion criteria: individual of both genders; age equal to or above 65 years of age; non-institutionalized individuals; they give permission to be part of the study and if the participant presents with a clinical condition or comorbidity, it must be stable and enable participation in aquatic-based exercises classes as approved by local medical staff. The exclusion criteria are: individuals less than 65 years of age; individuals with medical diagnosed pathologies that jeopardize their health while preforming aquatic based exercises, participants that attend less than 50% of all the sessions and participants who cannot complete all of the proposed tests.

The participants will be distributed randomly into different exercise groups based on their registration for one of the schedules offered for the hydro-gymnastics sessions. There will be three distinct schedules (9.00, 9.50 and 10.40), with the constraint being that the participants must attend the same schedule for the whole year. The 9.00 session includes the continuous aerobic program, the 9.50 session includes the interval aerobic program, and the 10.40 session includes the combined exercise program. This information will not be provided to participants before they register for the sessions so it will not be possible to establish any association between the schedule and the different exercises programs. Such information will be communicated later in the study. The control group will be randomly recruited from the community and will include those individuals that have not been involved in any kind of physical exercise during the last year.

Before each assessment moment, participants will be taken to a testing room, where the assessment tests will be performed. The room will be large and isolated, and the temperature will be controlled, and each assessment stages should be organized to provide maximum comfort and privacy to the participants during the tests. The research team will give the participants information about the tests they will perform for data collection, explain the purpose of each test, and explain the order and duration of the tests. Participants will be able to question researchers about any doubts that they may have regarding the tests and any possible consequences. During the assessment, participants may pause the evaluation and continue on another day if they feel very fatigued or if they are not able to complete all the tests at that time. In this situation, a new date will be scheduled to continue the tests.

### 2.3. Protocols

The percentage of adherence to the physical exercise programs, for each program, will be calculated considering the total attended sessions: (S × 100)/T, where “S” indicates the number of sessions that have been attended by the participant during the study and “T” indicates the total number (56) of physical exercise sessions. The participants’ attendance will be recorded in a database. If a participant has two consecutive absences, they will be contacted and given motivational reinforcement to incentivize them to resume their physical exercise sessions.

During the study period, all adverse effects or health problems attributed to the physical exercise sessions or evaluation tests will be reported. Parameters such as muscle pain, excessive fatigue and general pain will also be reported and inserted in a database. Exercise technicians and researchers will be responsible for data collection as well as for gathering and communicating all relevant data.

Three physical exercise programs will be conducted for a time period of 28 weeks, two times per week (non-consecutive days) and have the following common characteristics: all sessions will have a duration of 45 min, taking into account previous studies [10] that suggest sessions of this duration seem to be sufficient to provide changes in several parameters in the elderly population; they will be aquatic-based (the water level will be between 0.80 and 1.20 m with a temperature of approximately 32 °C); and they will be conducted to the rhythm of music (bpm) that can be adjusted to achieve the target HR. Sessions are divided into three parts: the initial part, main part and final part, with common exercises in the initial and final part of the three programs. The initial part or warm-up has a duration of 10 min and the purpose is to assist participants to adapt to the water environment, more specifically, for participants’ to acclimatize and prepare for muscular and metabolic stimulation. Simple aquatic-based exercises will be conducted, e.g., displacement and isolated movements. The exercises increase in complexity and intensity during this initial phase. The final part has a duration of 5 min. This part will be divided into two phases: return to calm (relaxation) where relaxing exercises are conducted with the purpose of returning the participants’ heart rate (HR) value to a resting level. The second phase is composed of stretching routines that stretch the most exercised muscle groups stimulated in the main part of the session and reduce the level of lactic acid and the occurrence of post-exercise pain. The main part is different in the three physical exercise programs and their characteristics are described in Table 1. All physical exercise programs will be planned and implemented according to the recommendations of the American College of Sports Medicine [19] and conducted by specialized physical exercise technicians (with a degree in sports science) with specialization in hydro-gymnastics (instructor course—level 1).

HR monitors (Polar V800) will be used on the participants during all sessions of the different physical exercise program. The intensity of each exercise program will be monitored using the data provided by the Polar V800 and adjusted accordingly. As a precaution and safety measure, the intensity will be indirectly calculated using the following equation [20]:(1)Target HR=((Maximum HR−Resting HR) % intensity)+Resting HR

*Maximum HR* is calculated with the following equation for senior populations [21]:(2)Maximum HR=207−(0.7×age)

The control group consists of non-institutionalized individual participants who have not partaken in any physical exercise during the preceding year. These participants will be encouraged to conduct their daily activities as usual, except for the data collection (M1 and M2) organized by the researchers.

### 2.4. Instruments

#### 2.4.1. Individual Characterization

In the first assessment moment (M1), all participants will fill in a clinical survey to help with the clinical characterization of each participant. This document includes the following information: civil status, regular medication, infections, allergies, diseases, annual doctor consultations, average hours of daily sleep, supplement use, latest blood panel, information on dietary habits, smoking habits and drug use.

#### 2.4.2. Environmental Characteristics

The physical exercise programs are aquatic-based, and will take place in water at temperatures that follow regional health guidelines (30–32 degrees Celsius) in the swimming pool complex of Piscina Municipal da Sertã (indoor pool), Portugal. During the study, daily monitoring of parameters such as the pool water temperature, free chlorine, combined chlorine, pH, relative humidity and pier external temperature will be conducted and the results will be inserted in a database by two facility staff members. An external certified company will conduct a bi-weekly analysis of the following parameters: pH, conductivity, free chlorine, total chlorine, temperature and, bacteriological tests (total germs, total coliforms, Escherichia coli, fecal enterococci, total staphylococci, coagulase-producing staphylococci, and pseudomonas aeruginosa).

#### 2.4.3. Anthropometry

Anthropometric measurements will be conducted by a certified investigator by FCDEF-UC. The following parameters will be evaluated: (i) stature, using a portable stadiometer, Seca Bodymeter^®^ (model 208, Hamburg, Germany) with a precision of 0.1 cm; (ii) weight, body mass index (BMI), visceral fat, percentage of fat and muscle mass using a portable scale from Seca^®^ (model 770, Hamburg, Germany) with 0.1 kg accuracy; and (iii) waist circumference, arms and legs using a retractable fiberglass tape (model Hoechst mass-Rollfix^®^, Sulzbach, Germany) with an accuracy of 0.1 cm.

#### 2.4.4. Physical Function

Physical function will be assessed using the Senior Fitness Test, developed and reviewed by Rikli and Jones [22] and validated for the Portuguese population [23]. It is composed of the following test items:Chair stand, assesses lower body strength and consists of the maximum number of full stands that can be concluded in 30 s. Necessary equipment: chair and stopwatch.Arm curl, assesses upper body strength and consists of the maximum number of bicep curls that can be completed in 30 s while holding a hand weight. Necessary equipment: 2.27 kg hand weight for women and 3.63 kg for men, chair and stopwatch.2-min step, assesses aerobic endurance and consists of maximum number of full steps completed in 2 min, a full step is recorded when each knee reaches the point midway between the patella (kneecap) and iliac crest (top hip bone). Necessary equipment: stopwatch, sticky-tape and ruler.Chair sit and reach, assesses lower body flexibility and is conducted from a sitting position where one of the participant’s legs is extended while the other is flexed and where hands are reaching towards the toes. This test is assessed in cm and is positive (+) if the extended fingers pass the tip of the toes or negative (−) if the extended fingers do not pass the tip of the toes. Necessary equipment: chair and ruler.Back scratch, assesses upper body flexibility and is conducted with one hand reaching over the shoulder in the direction of the floor and the other hand up the middle of the back in the direction of the head. This test is assessed in cm and is positive (+) if both hands overlap and is negative (−) if overlapping does not occur. Necessary equipment: ruler.Timed up and go, assesses agility and dynamic balance and is conducted from a starting sitting position where the participant stands up and walks, as fast as possible, to and from a distance 2.44 m (marked by a cone). Necessary equipment: chair, cone and stopwatch.Hand grip, assesses hand grip strength and consists of asking the participant to grip a dynamometer with maximum achievable force, the output value of the device is then registered. Necessary equipment: Lafayette hydraulic manual dynamometer (model J00105).

#### 2.4.5. Cognitive Function

Cognitive function will be assessed with the Portuguese version of the Mini Mental State Examination (MMSE) [24]. The MMSE, evaluates the following cognitive areas: orientation, short term memory, attention and calculation capacities, long term memory and language capabilities. The final score has a maximum of 30 points, and scores below 24 can be used as an aid in the assessment of dementia. The test will be used as an instrument to create a cognitive profile with the following criterium [25]: severe cognitive impairment (scores between 1 and 9 points); moderate cognitive impairment (scores between 10 and 18 points); mild impairment (score between 19 and 24 points), normal cognitive profile (scores between 25 and 30 points).

#### 2.4.6. Mental Health

Mental health will be assessed using the following scales and questionnaires validated for the Portuguese population: the Rosenberg Self-Esteem Scale (RSES) [26]; Physical Self-Perception Profile for Clinical Populations (CPSPP) [27]; World Health Organization Well-Being Index (WHO-5) [28]; Satisfaction With Life Scale (SWLS) [29]; EuroQol (EQ-5D) [30]; Geriatric Depression Scale (GDS) [31] and; Perceived Stress Scale (PSS) [32].

RSES, assesses global self-esteem and is composed of 10 items that are answered using a 4-point Likert scale, the answers vary from “I totally agree” to “I totally disagree”. In items 1, 2, 4, 6 and 7 the score is reversed. Global self-esteem is represented by the summation of all individual scores, providing a final score ranging from 10 to 40 points, where higher scores indicate higher self-esteem.CPSPP, is an instrument designed to provide a self-assessment summary of the physical characteristics of elderly groups in clinical and rehabilitation settings. A scale is defined by six subscales of three items that evaluate the following subdomains: functionality, physical health, sports competence, physical attractiveness, physical strength and physical self-worth. Answers to the items are displayed in an alternative structured format that is designed to eliminate social desirability bias. The score can vary between 3 to 12, with higher scores representing better performance.WHO-5, is an instrument that assesses psychological well-being. It is a self-administrated short questionnaire composed of 5 items with positive words, these words are related to a positive mood (good mood, relaxation), vitality (being active and waking up fresh and rested) and general interests. Each item is classified on a 5-point Likert scale, ranging from 0 (not present) to 5 (constantly present). The scores are summed, with the final score ranging from 0 to 25 points. The final score is then converted to a scale of 0 to 100 (by multiplying by 4), where higher scores represent a higher level of well-being and better quality of life. A final score equal to or below 50 points represents poor well-being but does not necessarily mean depression. A final score equal to or below 28 possibly indicates clinical depression.SWLS, assesses global cognitive parameters of life satisfaction. It is composed of 5 items with a 7-point Likert scale. The answers indicate the level of agreement the participant feels with each item. The final score ranges from 1 to 35 points, where higher final scores indicate higher satisfaction with life.EQ-5D, is an instrument that assesses general health status. It consists of two parts: the EQ-5D health descriptive system and the EQ visual analogue scale. The descriptive system consists of five dimensions (mobility, personal care, usual activities, pain/discomfort and anxiety/depression). The participant is asked to indicate their health status by selecting the most appropriate options in the five dimensions. The visual analogue is self-assessed and is conducted in a scale from the lowest rate (0) “the worst health you can imagine” to the highest rate (100) “the best health you can imagine”.GDS, assesses life satisfaction, interruptions in activities, annoyances, isolation, energy, joy and memory problems. It consists of fifteen easy to understand questions and has a binary answer system (0 or 1 point) for answers of “no” and “yes”, respectively. A participant who obtains a final score between 0 and 5 points is considered healthy; scores between 6 and 10 points indicate signs of mild to moderate depression; scores between 11 and 15 points indicate signs of severe depression.PSS, is an instrument to measure perceptions of stress. It is composed of 14 items, where 7 items are considered as positive aspects while the rest are considered as negative aspects. The questions are about feelings and thoughts during the last month. A point reversal is conducted on items 4, 5, 6, 7, 9, 10 and 13. The final score may vary between 14 and 70 points and a higher score indicates higher stress levels.

#### 2.4.7. Assessment of Carotid Arteries Intima-Media Thickness

The carotid arteries intima-media thickness assessment takes place with the participant lying down in a dorsal position. Then, the following parameters are evaluated using a sphygmomanometer from Riester (Model RI-championN^®^, Jungingen, Germany): heart rate (HR), systolic blood pressure (SP) and diastolic blood pressure (DP). The intima-media thickness of the right and left carotid arteries are measured with a Doppler two-dimensional ultrasound and are assessed with the AIRC study protocol [33]. The following values are then recorded through a portable ultrasound from General Electric^®^ (VIDe, Vancouver, Canada) with probe linear 11 L: Intima-media thickness (IMT); systolic diameter (SD), diastolic diameter (DD), peak systolic velocity (PSV) and end-diastolic velocity (EDV).

#### 2.4.8. Heart Rate Variability (HRV) Measurement

HRV will be assessed according to the procedures of Abad et al. [34] using Polar V800 heart rate monitors. Participants will place the sensor, which is synchronized with the V800 clock, on their chest beneath their pectoral muscles. Then, the participants will be asked to lie down in a dorsal position, in silence, with open eyes and with a calm respiration. The test will have a duration of 10 min in a calm, silent and low-light environment. After the conclusion of the test, HRV measurement data are downloaded from the Polar Flow Web Service.

#### 2.4.9. Biochemical Markers

Blood samples will be drawn via venipuncture from fasting participants. For each participant a total of 18.5 mL of blood will be drawn. Of the 18.5 mL, 3.5 mL will be used by the clinic to assess lipid panel values (HDL, LDL, glucose, triglycerides). The remaining 15 mL, processed by the college laboratory, will be divided into three tubes: two serum separator tubes and one ethylenediaminetetraacetic acid (EDTA) tube. Once the test tubes arrive at the university laboratory a complete blood count (CBC) will be conducted using an automatic hematology analyzer Coulter Act Diff, Beckman Coulter, USA. Next, the test tubes are centrifuged for10 min at 3500× *g* rotations per minute and stored in cryogenic test tubes. Levels of HbA1C, IL-1, IL-1ra, IL-6, IL-10, TNF-alpha, Adiponectin, Leptin, MIP-1alpha, MCP-1, SOD, MMP-9, are subsequently analyzed with ELISA Invitrogen^®^ CA kits (Bender MedSystems GmbH, Vienna, Austria).

### 2.5. Ethical Aspects

The researcher will be responsible for the data integrity and validity during the entire study. All data collected will be confidential and used exclusively for scientific purposes. Anonymization procedures will be conducted under the Data Protection Regulation of 25 May 2018. Data anonymization will be implemented by attributing a code to each participant. Each participant will receive a unique identification code that will correspond to their process; this code will be visually accessible with the use of “code cards”. These “code cards” will be used by the participants during data collection. Once the data collection is finalized the participants will be asked to destroy their “code cards”, thus finalizing the anonymization process.

Data will be stored in an Excel Microsoft Office 2016 database. Access to this database will be protected with a password and will be restricted to the main researcher responsible for the data collection. Data backups will be carried out regularly by the main researcher.

Participation in this study is purely voluntary, with no reprisals for non-participation. All subjects will be asked for their informed consent before they start participating in the study. The study will be conducted in accordance with the Declaration of Helsinki, and the protocol was approved by the Ethics Committee for Health of the Faculty of Sport Sciences and Physical Education of the University of Coimbra (reference: CE/FCDEF-UC/00462019).

The results from this study will be disseminated through publication in various scientific journals. The participants, if desired, will receive a copy of the main results and publications.

### 2.6. Statistical Analysis

The size and statistical power of the sample will be calculated using the G*Power software application [35]. The following parameters will be considered: F test (ANOVA); effect size: 0.25; α-level: 0.05; statistical power: 0.95; number of groups: 4; number of measures: 2 (pré and post intervention); a 30% margin for possible losses and refusals. Therefore, the initial size of the total sample was estimated at 76 participants.

The collected data will be subjected to descriptive statistical analysis where values such as maximum, minimum, mean and standard deviation will be calculated for each variable in each assessment moment. Afterwards, data normality will be tested by considering the response to three conditions: z-values from Skewness and Kurtosis tests; *p*-value from Shapiro-Wilk test; and visual inspection of generated histograms. Parametric data will be analyzed using the Student’s *t*-test for independent samples to compare the groups and two paired samples to compare the different moments (M1 and M2). Nonparametric data will be analyzed using the Mann–Whitney U test to compare the groups and the Wilcoxon test to compare the different moments (M1 and M2). Statistical analysis will be performed using the Statistical Package for the Social Sciences (SPSS) statistical software, version 25.0. The level of significance used will be *p* ≤ 0.05.

## 3. Expected Results/Discussion

The main aim of this study is to assess the impact of different aquatic exercise programs (a continuous aerobic program, aerobic interval program and combined program) on risk markers of cardiometabolic diseases in older people. The study is guided by scientific-based evidence on the practice of physical exercise in the elderly population [11,36].

The practice of regular physical exercise is considered as the optimum tool in the prevention and treatment of various types of cardiometabolic diseases. Thus, it is important that research on this topic continues; by doing so, new intervention methods can be identified and their efficacy validated. The offer of aquatic physical exercise programs is highly successful among the elderly because they transform physical exercise into something more pleasant and suitable for this particular population. However, the high costs of maintaining aquatic environments and the additional difficulties associated with assessment methodologies and specific equipment needed to monitor exercise in aquatic environments, means this topic not yet been sufficiently explored in the literature.

After completion of the data collection and analysis, the participants in the experimental groups are expected to show positive developments with regard to the anthropometric level, physical function, the intima-media thickness of the carotid arteries, heart rate variability, cognitive function, mental health and a number of biochemical markers. It is also expected that statistically significant differences will be found between the exercise groups for some of the variables. In the control group, no changes are expected in the analyzed variables. It is believed that the expected results can be attributed to the physical and physiological effects of the aquatic environment associated with the different proposed exercise protocols. The results will be published after the study is completed.

## Figures and Tables

**Table 1 ijerph-17-08678-t001:** Characteristics of the three physical exercise programs applied for 28 weeks (continuous aerobic, interval aerobic and combined).

Program	Description	Intensity (Week 1–13)	Intensity (Week 14–28)	Exercises
Continuous Aerobic	30 min exercise aerobic (moderate intensity)	60–65% maximum HR	65–70% maximum HR	Basic hydro-gymnastics exercise, with some variations: running, bounce, kicking, pendulum jumping, skiing, twister and horse.
Interval Aerobic	10 min exercise aerobic (moderate intensity)	60–65% maximum HR	65–70% maximum HR	Basic hydro-gymnastics exercise, with some variations: running, bounce, kicking, pendulum jumping, skiing, twister and horse.
	5 min exercise aerobic (high intensity)	70–75% maximum HR	75–80% maximum HR	
	10 min exercise aerobic (moderate intensity)	60–65% maximum HR	65–70% maximum HR	
	5 min exercise aerobic (high intensity)	70–75% maximum HR	75–80% maximum HR	
Combined	15 min exercise aerobic (moderate intensity)	60–65% maximum HR	65–70% maximum HR	Basic hydro-gymnastics exercise, with some variations: running, bounce, kicking, pendulum jumping, skiing, twister and horse.
	15 min muscular strengthening exercises	2 steps 12 repetitions	3 steps 16 repetitions	Exercises with auxiliary equipment (dumbbells, pool noodles, etc.): elbow extension/flexion; shoulder extension/flexion; shoulder abduction/adduction; hip abduction/adduction; hip flexion/extension; knee flexion/extension; dorsal and plantar flexion of the ankle.

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
