# Peer review of "Impact of Aquatic-Based Physical Exercise Programs on Risk Markers of Cardiometabolic Diseases in Older People: A Study Protocol for Randomized-Controlled Trials"

_ijerph, 2020, doi:10.3390/ijerph17228678_

Round 1
Reviewer 1 Report
Dear Authors,
Thank you for this interesting study protocol.
The main purpose of this study protocol is to assess the impact of different aquatic-based physical exercise programs on markers related to cardiometabolic diseases in old people.
Please, find some minor comments below.
Methods:
Please, add more description of the exercise types in each group. I think it will be better if it presented as a table.
2.4. Size and statistical power of sample
The size and statistical power of the sample will be calculated using the G*Power software
application (54, 55).
All questionnaires should be validated for your sample. Please, add it as a step in the methods section.
Please, add some description of the trainers' qualifications.
Please, calculate the sample and add it to the protocol!
Line 155. Anthropometric measurements will be conducted by a certified investigator.....
What does it mean? certified from the university or ministry of health or other institution?
Statistical Section: should be as a Planned Statistical Analysis
Discussion section:
This is only a study protocol so, there are no results or discussion to present. Please, the discussion section must be modified as it is very general. Please, write it as Expected Results/Discussion and do it briefly.
Author Response
Point 1: Please, add more description of the exercise types in each group. I think it will be better if it presented as a table.
Response 1: More descriptions of physical exercise programs have been added (line: 131-138).
Point 2: All questionnaires should be validated for your sample. Please, add it as a step in the methods section.
Response 2: References for the validation of all questionnaires were added to the study sample (line 253-257).
Point 3: Please, add some description of the trainers' qualifications.
Response 3: Descriptions of the trainer's qualifications were added (line: 149-152).
Point 4: Please, calculate the sample and add it to the protocol!
Response 4: The sample size was calculated and added to the protocol (line:348-352).
Point 5: Anthropometric measurements will be conducted by a certified investigator.....What does it mean? certified from the university or ministry of health or other institution?
Response 5: Anthropometric measurements will be conducted by a certified investigator through the FCDEF-UC (line: 209).
Point 6: Statistical Section: should be as a Planned Statistical Analysis.
Response 6: The statistical section has been reformulated (line: 348-362).
Point 7: This is only a study protocol so, there are no results or discussion to present. Please, the discussion section must be modified as it is very general. Please, write it as Expected Results/Discussion and do it briefly.
Response 7: The discussion section has been reformulated (line: 364-384).
Reviewer 2 Report
The subject addressed by the authors of the manuscript is of interest, however, it must be reconsidered after an important review (results are missing), the references are excessive and 32% are outdated (more than 5 years), which limits the investigation. Finally, it is necessary to carefully order the methodology section, since very dispersed and redundant information is presented.
Here are my observations:
objective
This is not clear, since it is mentioned 3 times throughout the text, and there are also differences in the writing of this, so it must be unified in the summary and at the end of the introduction and taken from the methodology.
- In the abstract, the objective says: "The main purpose of the present research study is to understand the impact of different aquatic-based physical exercise programs on risk markers of cardiometabolic diseases." (Lines 21-22-23).
- At the end of the introduction the objective reads: The present investigation and its specific experimental design using different types of aquatic-based exercise programs aims to assess the impact of such exercise programs on risk markers of cardiometabolic diseases in old people, and provide evidence for the predictable positive effects of its use on health and well-being of old populations. (Lines 66-67-68)
- In methodology, the objective is mentioned again and says: The main purpose of this study is to assess the impact of different aquatic-based physical exercise programs on markers related to cardiometabolic diseases in old people. (Lines 66-67-68).
Materials and Methods
Structuring the methodology to better deliver the information, since in my opinion it is very messy, it is suggested to structure by design, participants, protocols, instruments, ethical aspects and statistical analysis (include the size of the effect), avoiding including too many subtitles.
Line 81 says: and conducted by specialized physical exercise technicians with specialization in hydrogmnastics. The degree of specialization, years dedicated to teaching, certifications obtained, etc. should be mentioned.
Lines 105 to 109. This should go along with the statistical analysis section.
Results
The results are not presented.
Discussion
In order to evaluate the discussion in a good way, the results are required.
conclusion
This does not respond to the research objective, it only gives a series of suggestions and / or recommendations. Therefore, it is suggested to add recommendations after the discussion, and write the conclusion that responds to the objective of the study. Which of the three programs is the most effective?
References
Some references should be updated, in addition to selecting a reasonable number, and checking that they are in accordance with the journal's publication standards.

Author Response
Point 1: This is not clear, since it is mentioned 3 times throughout the text, and there are also differences in the writing of this, so it must be unified in the summary and at the end of the introduction and taken from the methodology.
Response 1:The objective was unified and removed from the methodology section.
Point 2: Structuring the methodology to better deliver the information, since in my opinion, it is very messy, it is suggested to structure by design, participants, protocols, instruments, ethical aspects and statistical analysis (include the size of the effect), avoiding including too many subtitles.
Response 2: The methodology was structured according to your recommendations (line: 69-362).
Point 3: Line 81 says: and conducted by specialized physical exercise technicians with specialization in hydrogmnastics. The degree of specialization, years dedicated to teaching, certifications obtained, etc. should be mentioned.
Response 3: Descriptions of the exercise technicians qualifications were added (line: 149-152).
Point 4: Lines 105 to 109. This should go along with the statistical analysis section.
Response 4: The statistical section has been reformulated (line: 348-362).
Point 5: The results are not presented.
Response 5: The article is a study protocol. there are still no results to display.
Point 6: In order to evaluate the discussion in a good way, the results are required.
Response 6: The discussion section has been reformulated (line: 364-384).
Point 7: This does not respond to the research objective, it only gives a series of suggestions and / or recommendations. Therefore, it is suggested to add recommendations after the discussion, and write the conclusion that responds to the objective of the study. Which of the three programs is the most effective?
Response 7: It is still not possible to know which exercise program is the most effective.
Point 8: Some references should be updated, in addition to selecting a reasonable number, and checking that they are in accordance with the journal's publication standards.
Response 8: The reference section has been reformulated (line: 395-480).
Round 2
Reviewer 2 Report
No comment
This manuscript is a resubmission of an earlier submission. The following is a list of the peer review reports and author responses from that submission.
Round 1
Reviewer 1 Report
Overall, the reason for carrying out this study is very vague and unclear. The authors measure a wide variety of outcomes without providing proper reasoning, validation of tests, or hypotheses. The entire paper needs to be more focused. This reads like a first-draft that should be reviewed by the co-authors, and not the journal reviewers.
Introduction
Almost 50% of the introduction discusses the detrimental effects of a sedentary lifestyle. Yet, sedentariness/sedentary behavior is not the same as physical inactivity (which was in the thesis statement of the first paragraph). Do the authors aim to use the exercise program to counteract sedentary behavior or physical inactivity? This is unclear in the introduction.
The third paragraph (lines 57-60) is very vague. Which specific changes occur as a result of physical exercise and which study reported these results? The authors shouldn’t clump all citations at the end of the paragraph.
Line 65-66: “specific aquatic exercise programs so popular and successful among older participants” – any citation or reasoning for this? The reasons for why the authors plan to study aquatic exercise programs need to be elaborated more.
2.3 Study Participants and Recruitment
Line 96: “individuals of both genders” is not specific and does not use language of inclusivity and respect. Do the authors mean biological sex?
How do the authors measure “independent” and “active” when recruiting participants?
The authors describe the recruitment area and inclusion/exclusion criteria. The recruitment process was not described. Did the authors recruit from telephone calls? Medical hospitals? How do the authors ensure generalizability of results from the recruited participants?
How was the control group recruited?
2.4 Size and Statistical Power of Sample
How is the effect size determined?
Please provide citation for G*Power software.
2.5 Study randomization
This process cannot be termed “randomization” as the participants self-select a group that they would like to participate in. The participants are blinded to the allocation, but the process is not randomized, e.g., by using computer generated random numbers.
The authors should test for potential baseline imbalances between all four groups using all demographical information (section 2.8.1. Individual Characterization) and all the suggested parameters. This should be reported in this section or in Section 2.10.2 Statistical Analysis.
2.6 Familiarization and Reliability
Please specify exactly what the controlled temperature is, and reason for that specific temperature.
How would the intensity of the warmup be controlled for? Why is a warmup needed before assessment?
2.8 Assessment Instruments
The authors need to provide reasons for measuring all the suggested parameters (i.e., anthropometry, physical functions, psychological health, cognitive level, carotid arteries intima media thickness, heart rate variability, and biochemical markers). Furthermore, the authors need to provide reasons for each test within each suggested parameter. Have these tests been validated? Do the authors have any hypotheses to these tests with regards to the exercise program?
2.8.2. Ambiental Characteristics
Are the swimming pools indoor or outdoor?
2.8.6 Psychological Health
How is psychological health defined? Why is cognitive “level” (cognitive “performance” or “function” is usually used, instead of “level”) not categorized under psychological health?
The CPSPP does not appear to measure psychological health.
2.9.1. Adherence to Physical Exercise Programmes
Participants who need motivational reinforcement need to reported in the results later.
Please describe the motivational reinforcement.
2.9.2. Physical Exercise Programmes
Why are there common exercises? What does it mean by “common”?
2.10.2. Statistical Analysis
Why was the Shapiro-Wilk test selected?
Exactly which statistical tests will be used? For example, which type of ANOVA, regression, or multilevel analysis will be used? And why?
Will the alpha-level be one-sided or two-sided?
- Discussion
The authors need to distinguish between sedentary behavior, physical activity, physical inactivity, and exercise.
The discussion is basically the reasoning for why these parameters were measured in the study. These should be introduced either in the introduction or methods section. The discussion section should discuss concerns regarding the expected outcome, implications of outcomes, strengths, weaknesses, future direction, and so on.
Reviewer 2 Report
Dear Author,
Thank you for this interesting paper.
The main purpose of this study is to assess the impact of different aquatic-based physical exercise programmes on markers related to cardiometabolic diseases in old people.
Please find some minor comments below.
Abstract:
Please add the sample in the abstract, how many subjects, in addition to the main anthropometric measurements.
Introduction:
The introduction is short and needs further studies
Methods:
- Can the researchers explain the consequences of collecting data for each participant?
At the first evaluation moment, for example:
- There are many variants such as regular medications, infection, allergies, diseases, annual physician consultations, average daily sleep hours, use of supplement consumption, the latest blood plate, information on dietary habits, smoking habits, and drug use. How was the excrement or training design affected by these variables?
- L 147. During this study a daily monitorization of parameters such as pool water temperature. have you asked the subjects about the water temperature or other variables when possible?
- It may be different based on body style; this may affect their response and participation in training.
- L 153. How much time needed to finish the pre- measurements, all those questionnaires, assessment of carotid arteries intima-media thickness, Heart Rate Variability (HRV) Measurement, and Biochemical Markers for each subject and how many one daily did you measure? It seems more than 4-6 hours for each one!!
- Why the session duration was 45 minutes? Is there a reference for this point or it is only suggestion from the authors? Is it a appropriate for your subjects in this age?
- 284 part of the three programmes. Where are these programs, I need to know what kind of exercises you apply? For me there is no training program it is only a description for the intensity and the design!!
- How did you measure HR during the session?
- How did you measure HR during the session?
- Line 321. Did you measure each one during the training session by Polar V800? Or you asked the subjects to tell you how many HR they have?
- How did you target the HR to be for example 130 BPMs while each subject in this age has a different heart rate at rest?
Results
- Where is the results section?
Dissection:
Some paragraphs for example line 367-390 looks like introduction and literature reviews not suitable for the dissection section.